# Stem-Maps of Forest Restoration Cuttings in *Pinus ponderosa*-Dominated Forests in the Interior West, USA

**Justin P. Ziegler [1],\*, Chad M. Hoffman [1], Mike A. Battaglia [2] and William Mell [3]**

[1] Department of Forest & Rangeland Stewardship, Colorado State University, Fort Collins, CO 80523, USA; c.hoffman@colostate.edu
[2] Rocky Mountain Research Station, US Forest Service, Fort Collins, CO 80526, USA; mbattaglia@fs.fed.us
[3] Pacific Wildland Fire Sciences Lab, US Forest Service, Seattle, WA 98103, USA; wemell@fs.fed.us
\* Correspondence: justin.ziegler@colostate.edu

**Abstract:** Stem-maps, maps of tree locations with optional associated measurements, are increasingly being used for ecological study in forest and plant sciences. Analyses of stem-map data have led to greater scientific understanding and improved forest management. However, availability of these data for reuse remains limited. We present a description of eight 4-ha stem-maps used in four prior research studies. These stem-maps contain locations and associated measurements of residual trees and stumps measured after forest restoration cuttings in Colorado, Arizona, and New Mexico. Data are published in two file formats to facilitate reuse.

**Dataset:** available as Supplementary Data

**Dataset License:** CC-BY-NC-SA

**Keywords:** ponderosa pine; thinning; dry forest; tree map; point pattern; spatial relationships

---

## 1. Summary

For over a century, it has been recognized that the properties and dynamics of ecological systems arise from the cumulation of adaptive behaviors of, and changes to, individual organisms [1,2]. As a corollary, ecologists can take a "plants-eye" view, using the spatial relationships between individuals to infer ecological processes and interactions [3,4]. However, rigorous analyses require fine-scaled detailed maps of individuals, the availability of which has historically lagged behind spatial statistical advances [3]. Today, the affordability and availability of technologies such as laser rangefinders, light detection and ranging (LiDAR) and geospatial information systems software as well as statistical software has led to a growing capacity for data collection and analysis. As a result, collecting and analyzing maps of individuals is now a common methodology in the ecologist's toolbox [3,5].

In context of forestry-related research, maps of individual trees or other types of woody vegetation are termed stem-maps. Stem-maps are an enumerated set of 2-tuples describing the locations of trees within a bounded two-dimensional space. Locations can be geocoded or arbitrary Cartesian coordinates, and locations can have associated attributes.

The collection of stem-maps has led to recent paradigmatic shifts. For example, once a sufficient corpus of research using stem-maps had been published, Larson and Churchill's [6] meta-analysis concluded that spatial heterogeneity was a hallmark of historic pre-settlement forests in the western US. This recognition is particularly insightful for management in drier forests with a once-frequent fire regime. In many of these forests, restoration is a common ecological objective as a century of fire suppression efforts has often yielded forest conditions which significantly deviate from those of historic forests [7]. Made aware of the

ecological significance of tree spatial patterning by Larson and Churchill [6], forest managers are now likely to incorporate spatially-explicit cutting guidelines into forest restoration plans.

The dataset presented here includes a series of stem-maps collected to evaluate forest restoration cuttings in ponderosa pine (*Pinus ponderosa* ex. Lawson) forests of the interior West, USA. This dataset was first described by us in Ziegler et al. [8]. Firstly, forest restoration in these dry forests was aimed at creating aggregated tree spatial patterns comprised of a mosaic of small tree groups and isolated trees [9,10]. Secondly, the cuttings aimed to reduce potential wildland fire behavior [9,10]. In Ziegler et al. [8], we used the dataset presented here to conduct point pattern analyses to compare the spatial patterns of trees before and after restoration cutting. We then used the individual tree locations and their individual geometries as inputs to a physics-based fire behavior model. The Wildland-urban interface Fire Dynamics Simulator (WFDS) model is interoperable with stem-map data, as WFDS resolves the distribution of vegetative fuel throughout three-dimensional space [11]. The dual assessment of tree pattern manipulation and spatial tree–fire behavior interactions is important considering how much remains unknown regarding how tree spatial arrangements alter wildland fire behavior [6,12,13]. Stem-maps are therefore useful data products for forest and wildland fire research [13].

Year after year, a greater proportion of forestry and ecology research articles are collecting data to investigate spatial forest structure at fine scales [5]. Yet, very few of these datasets are publicly available, retarding the potential rate of knowledge acquisition [13]. This disparity motivated us to describe and publish our dataset from Ziegler et al. [8]. This dataset has been used to address additional research questions. First, Hoffman et al. [14] used results from Ziegler et al. [8] in a meta-analysis to verify WFDS fire behavior predictions. Second, Tinkham [15] used a subset of this dataset to produce visualizations to aid forest managers planning to implement restoration cuttings. Last, Matonis et al. [16] supplemented the stem-map dataset to investigate relationships between overstory stocking and understory plant community composition. The potential reuse of this dataset has not yet been exhausted. Stem-maps of real-world plants can be used to investigate inter- and intra-specific interactions [17], hypothetical scenarios such as plant pest vulnerability [18], and may be used to inform silvicultural prescription development [19]. Thus far, results from the growing body of research using stem-maps have directly informed recommendations for forest management [9,10] and the development of forest management software [20].

## 2. Data Description

We provided data, in wide format, for each of the stem-maps as comma-separated values (csv) files. Each csv file contained a single stem-map with a single header line. Table 1 provides a description of the dataset's header fields. We also provided data in an R (R Core Team, Vienna, Austria) data file. The data file contained a tibble named "stemmaps" Tibbles are a class structured as a data frame. A data frame is a heterogeneous, two-dimensional array with named columns; data frames are heterogenous meaning columns can be of any data type (i.e., numeric, character, logical, or complex). The tibble had two columns; one contained the study site names and the other contained tibbles, each with the study site's stem-map, formatted as in Table 1. Missing values were represented with NA. Observations on stumps were limited to type, species, and diameter at stump height. For context, Table 2 and Figure 1 describe the forest structure represented by this dataset.

**Table 1.** Dataset fields describing stem-maps of forest restoration cuttings.

| Variable | Description |
| --- | --- |
| type * | Either tree or stump |
| species | four-letter code for tree species name (PLANTS Database; http://plants.usda.gov) |
| dbh | Diameter at breast height; tree bole diameter at 1.37 m above the ground |
| dsh | Diameter at stump height; tree bole diameter at 15 cm above the ground |
| ht | Tree height (m) |

**Table 1.** *Cont.*

| Variable | Description |
|---|---|
| cbh | Crown base height (m); compacted crown base height (Toney and Reeves, 2009) |
| cw | Crown width (m); average width of tree crown |
| x | Coordinate in the *x* dimension |
| y | Coordinate in the *y* dimension |

\* Stumps have values for only type, species, *x*, and *y*.

**Table 2.** Summary of forest restoration stem-maps.

| Type/Variable | Site | | | | | | | |
|---|---|---|---|---|---|---|---|---|
| | Bluewater | Dowdy Lake | Long John | Lookout Canyon | Messenger Gulch II | PA5 | Phantom Creek I | Unc Mesa |
| *Stump* | | | | | | | | |
| Mean dsh (cm) | 20.77 | 16.37 | 20.29 | 19.53 | 22.02 | 25.60 | 19.17 | 28.43 |
| *n* | 1055 | 1331 | 2248 | 1137 | 1663 | 495 | 2308 | 759 |
| *Tree* | | | | | | | | |
| Mean dbh (cm) | 30.39 | 21.45 | 22.22 | 30.91 | 17.54 | 24.00 | 12.77 | 19.59 |
| Mean dsh (cm) | 36.46 | 26.80 | 28.32 | 40.75 | 23.11 | 29.32 | 16.02 | 23.02 |
| Mean height (m) | 12.63 | 8.82 | 12.72 | 15.93 | 8.67 | 7.97 | 7.55 | 9.69 |
| Mean cbh (m) | 4.56 | 3.4 | 5.23 | 6.69 | 3.58 | 3.70 | 3.53 | 4.28 |
| Mean crown width (m) | 4.04 | 3.35 | 3.60 | 4.35 | 2.81 | 3.70 | 2.51 | 3.13 |
| *n* | 251 | 725 | 1071 | 812 | 1077 | 1177 | 1429 | 1247 |

dsh is diameter at stump height; dbh is diameter at breast height, and cbh is crown base height.

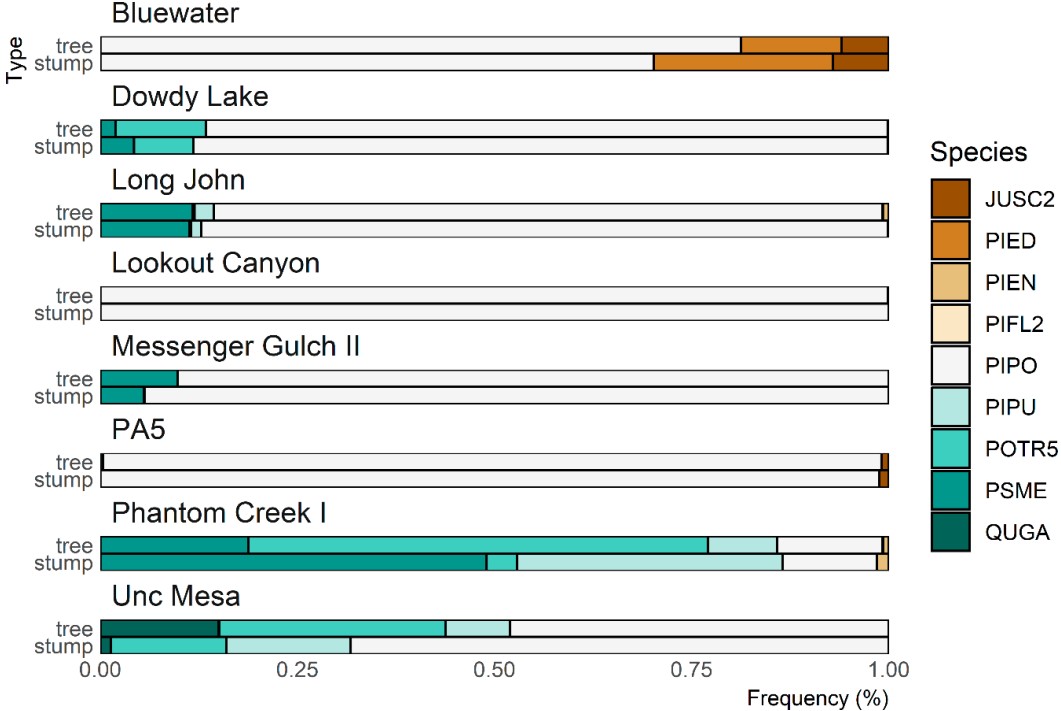

**Figure 1.** Species composition of stumps and trees across forest restoration cutting stem-maps; species names are coded according to the PLANTS database (http://plants.usda.gov).

## 3. Methods

This dataset contained eight stem-maps (Figure 2). All but one stem-map were reported in Ziegler et al. [8] and were measured between June and early August of 2012 and 2013. In August 2014, we measured an additional stem-map, Long John, and added this to the dataset. Five of these

stem-maps were located on the Front Range of Colorado, a sixth (Unc Mesa) was on the Uncompahgre Plateau of western Colorado, a seventh (Bluewater) was in the Zuni Mountains of western New Mexico, and the eighth (Lookout Canyon) was on the Kaibab Plateau of northern Arizona (Figure 3).

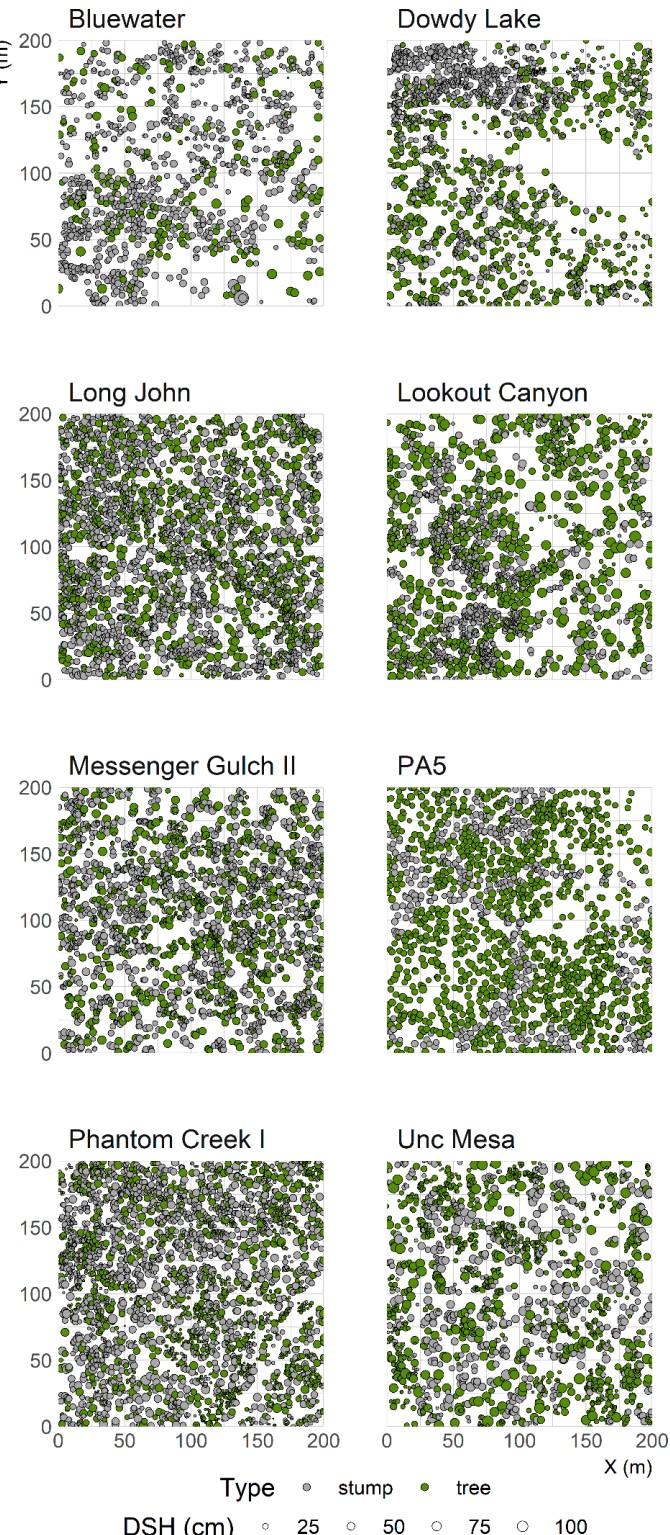

**Figure 2.** Stem-maps arranged by study site, which consisted of measured stumps and trees; points are sized by diameter at stump height (dsh; cm) and are not to scale. A 25 m × 25 m grid underlying the tree locations was used to facilitate stem-mapping.

We used an opportunistic sampling design and asked forest managers for study areas that met the following criteria:

- Cover type: Ponderosa pine forest is dominant.
- Topography: An average slope of less than 5% and few topographic features such as ridges or outcroppings.
- Representativeness: Cuttings should reflect the practices of restoration silviculture implementation at the time.
- Recency: Cuttings should have been implemented in the last decade, to limit observing post-cutting tree recruitment.

The forest managers then supplied us with locations of cutting units in which we placed a single 200 m × 200 m plot.

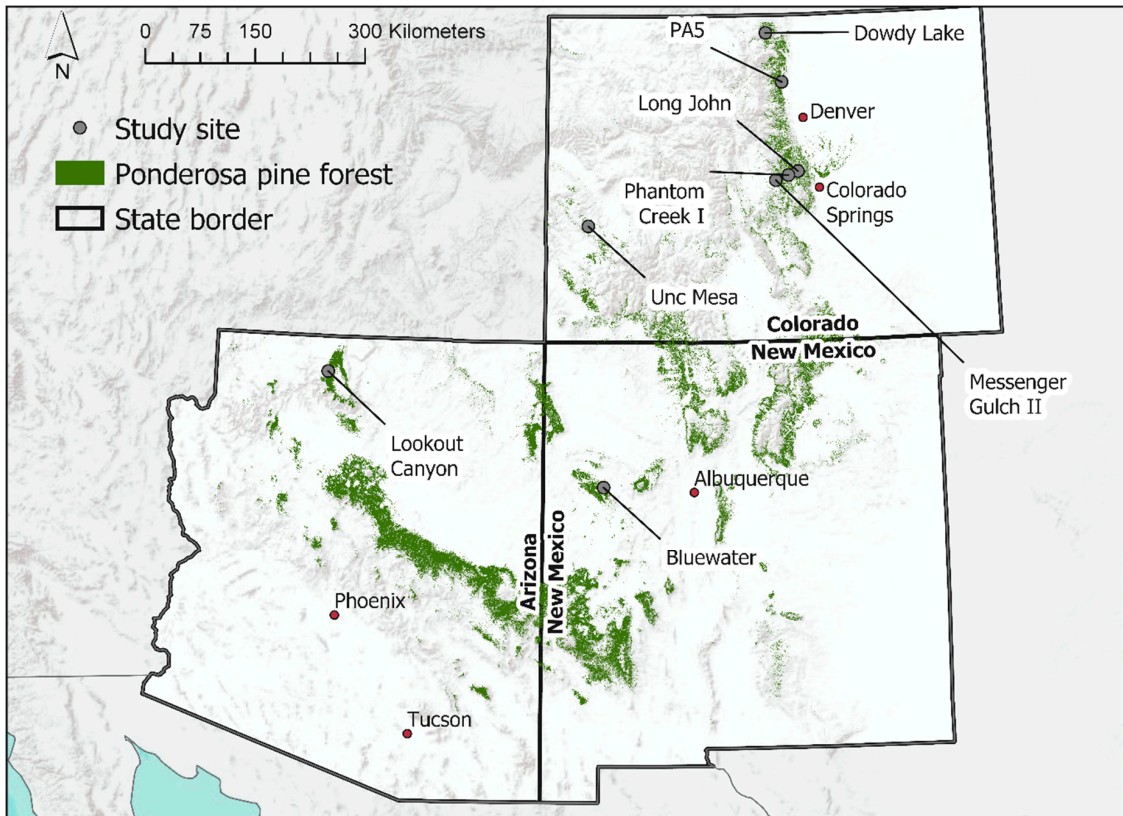

**Figure 3.** Geographic locations of stem-mapped study sites in southwestern USA; extents of ponderosa pine forests are added for context (Gap Analysis Project https://www.usgs.gov/core-science-systems/science-analytics-and-synthesis/gap/, accessed 04/07/2019).

In each plot, we first established a southwestern origin and measured its projected geographic coordinates (UTM Zone 12N in western plots and 13N in the Front Range) with an eTrex 20 handheld GPS unit (Garmin, Schaffhausen, Switzerland) use the Garmin® eTrex units can have a positional error of up to 12 m, we opted to map trees and stumps on a Cartesian coordinate system rather than record geographic coordinates. We used the southwestern origin as our anchoring point, and then used LaserAce 1000 laser rangefinders (Trimble, Sunnyvale, CA, USA) to divide the plot into four quadrants of 100 m × 100 m each. The rangefinders had an inbuilt inclinometer to ensure that ranged distances accounted for topographic slope. The rangefinders' lasers had a distance accuracy of 0.1 m, a compass heading accuracy of 2°, and an inclinometer accuracy of 2°. Once the approximate corners of quadrants were located, we used three rangefinders to adjust corner locations to minimize positional

error. We simultaneously measured the azimuth and distance from three positions to each corner in turn to minimize disagreement between individual rangefinders. This also ensured that quadrants maintained a square rather than a rhomboid shape. After the quadrants were laid out, we established a finer grid by dividing the quadrants into 25 m × 25 m cells. We used the same process of averaging the grid's intersections with multiple rangefinders. Once each single grid cell was surveyed, its boundaries were considered immutable.

Within each cell, we used the rangefinder to measure the azimuth and distance from the southwestern grid intersection to each tree (at least 1.37 m in height) and tree stump. We applied trigonometric relationships to determine $(x,y)$ tree locations:

$$(x, y) = (d \cos(\theta), d \sin(\theta)) + (x_i, y_i), \tag{1}$$

where $d$ is the distance from the grid intersection to the tree, $\theta$ is the azimuth in radians, and $(x_i, y_i)$ are the grid intersection's coordinates.

We recorded the species, diameter at breast height, diameter at stump height, tree height, compacted crown base height [21], and crown width for each tree. Tree height and compacted crown base height were measured with the rangefinder; other measurements were taken with a logger's tape. As the measurements were taken after the cuttings, we mapped stumps of cut trees and measured species and diameter at stump height. While we had reconstructed the full suite of tree measurements of cut stumps using linear regression [8], here, we archived only the observed data.

Quality control occurred in three stages. In the field, after all the tree and stump measurements within a grid cell were recorded onto paper forms, a reviewer checked that observations were sensical. Later, a reviewer compared digitized data against paper records, and made corrections for any errors during data entry. For the records that may have escaped quality control into the final stage, we examined the univariate distributions of individual tree and stump attributes. We found no instances of unlikely values at the upper end of attributes' ranges. At the low end, however, there were some instances in which observations were erroneously recorded with a value of 0. These observed values were changed to NA. We did not otherwise impute or modify any recorded values. In total, there were 12, 5, 2, 1, and 2 individual trees with a missing value for crown width, diameter at stump height, diameter at breast height, tree height, and crown base height, respectively.

**Supplementary Materials:** The following are available online at http://www.mdpi.com/2306-5729/4/2/68/s1.

**Author Contributions:** Conceptualization, J.P.Z., C.M.H., and M.A.B.; methodology, J.P.Z.; formal analysis, J.P.Z., C.M.H., and M.A.B.; investigation, J.P.Z.; resources, J.P.Z., C.M.H., M.B., and W.M.; data curation, J.P.Z.; writing—original draft preparation, J.P.Z.; writing—review and editing, J.P.Z., C.M.H., W.M.; visualization, J.P.Z.; supervision, C.M.H., M.A.B., and W.M.; project administration, C.M.H., M.A.B.; funding acquisition, C.M.H., M.A.B., and W.M.

**Funding:** Funding assistance came from the USDA Forest Service Rocky Mountain Research Station, the National Fire Plan, the Colorado State University Department of Forest & Rangeland Stewardship, and the Joint Fire Science Program (Project #13-01-04-53).

**Acknowledgments:** Assistance in data collection was provided by Megan Matonis, Tony Bova, Will Grimsley, Larry Huseman, Don Slusher, Emma Vakili, and the Colorado Forest Restoration Institute. Forest managers, Dick Edwards, Chad Julian, Matt Reidy, Matt Tuten, Jeff Underhill and Jim Youtz aided in identifying field sites.

**Conflicts of Interest:** The authors declare no conflict of interest.

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
