# Peer review of "Stem-Maps of Forest Restoration Cuttings in Pinus ponderosa-Dominated Forests in the Interior West, USA"

_data, 2019_

Reviewer 1 Report

The authors provide spatial-explicit data for five plots able to characterize the situation of about 1000 thousand trees and stems per plot. The data is very valuable and will be used for other researchers to advance the understanding on how spatial distribution can influence ecological processes in these forests. Therefore, I am recommending accepting this data manuscript with only two minor suggestions  for the authors:

- The process to ensure quality of the data has not been adequately described. What did the authors do when they found inaccuracies, wrong data, etc. Do they re-measure the tree position an atrributes, remove it from the data base, etc.?

- For international readers not familiar with North American geography, Figure 3 should include some easily recognizable items such as state names, mountain chains names, location of major cities, etc.

Author Response

Authors: Dear reviewer, thank you for taking the time to review our manuscript. We believe we have satisfied your concerns and your suggestions will ultimately improve the quality of the manuscript.

Reviewer: The authors provide spatial-explicit data for five plots able to characterize the situation of about 1000 thousand trees and stems per plot. The data is very valuable and will be used for other researchers to advance the understanding on how spatial distribution can influence ecological processes in these forests. Therefore, I am recommending accepting this data manuscript with only two minor suggestions  for the authors:

- The process to ensure quality of the data has not been adequately described. What did the authors do when they found inaccuracies, wrong data, etc. Do they re-measure the tree position an atrributes, remove it from the data base, etc.?

Authors: We appreciate that the reviewer (and likely readers) will be interested in data quality. We have added additional detail which should satisfy readers.

“Quality control occurred in three stages. In the field, after all the tree and stump measurements within a grid cell were recorded onto paper forms, a reviewer checked that observations were sensical. Later, a reviewer compared digitized data against paper records, correcting for any errors during data entry. Some records may have escaped quality control into the final stage of quality control; in response, we examined the univariate distributions of individual tree and stump attributes. We found no instances of unlikely values at the upper end of attributes’ ranges. At the low end however, there were some instances in which observations were erroneously recorded with a value of 0. These observed values were changed to NA. We did not otherwise impute or modify any recorded values. In total there were 12, 5, 2, 1, and 2 individual trees with a missing value for crown width, diameter at stump height, diameter at breast height, tree height, and crown base height respectively.”

Reviewer: - For international readers not familiar with North American geography, Figure 3 should include some easily recognizable items such as state names, mountain chains names, location of major cities, etc.

Authors: We thank you for catching this oversight. We have added state names and labelled major city locations to Figure 3. We also clarified in the figure caption that the map extent was focused on the southwestern US.

Reviewer 2 Report

Main comments:

The authors seem to be very precise in their description of the purpose and measurement of the stem maps. The only weak part of this paper involves lines 120-122. Over the last 15 years, several peer-reviewed papers have been published on the positional accuracy of recreation-grade GPS receivers (the class of GPS receiver in which the Garmin eTrex falls) at GPS test courses throughout the United States. On average, through these studies, the horizontal positional accuracy has been stated to be 6-10 m. In these studies it was rare to even have a single waypoint fall within centimeters of their true positions. So, it would seem necessary that the authors describe how closely the GPS data actually described the four corners, which were expected to represent a square 200 m to each side. Since a rangefinder was then likely employed from each corner to measure the locations of trees, the GPS positional error likely propagated throughout the stem mapped data.

Minor comments:

Line 43: "a century of fire cessation" is probably too strong, absolute, and misleading. Fires occurred. It has been a century of actively controlling and stopping wildfires.

Line 45: Again, a statement probably too absolute ..."some forest managers now incorporate spatially..."

Lines 68-71: Besides the ecological research uses of stem maps, they can also be used in planning uneven-aged harvests, as is suggested in this Forests paper: https://doi.org/10.3390/f6041121.

Line 78: define "tibble".

Line 153: Typo in the paper title.

Line 156: What is "Cop." ?

Line 192: Inconsistent representation of authors' names.

Author Response

Reviewer: The authors seem to be very precise in their description of the purpose and measurement of the stem maps. The only weak part of this paper involves lines 120-122. Over the last 15 years, several peer-reviewed papers have been published on the positional accuracy of recreation-grade GPS receivers (the class of GPS receiver in which the Garmin eTrex falls) at GPS test courses throughout the United States. On average, through these studies, the horizontal positional accuracy has been stated to be 6-10 m. In these studies it was rare to even have a single waypoint fall within centimeters of their true positions. So, it would seem necessary that the authors describe how closely the GPS data actually described the four corners, which were expected to represent a square 200 m to each side. Since a rangefinder was then likely employed from each corner to measure the locations of trees, the GPS positional error likely propagated throughout the stem mapped data.

 Authors: Thank you for your review. Your comments have helped us improve the quality of the paper.

With regards to your comment regarding GPS accuracy, we had significantly simplified our description for the sake of brevity but ultimately led to an inaccurate description. Indeed, we found that surveying the plot with a rangefinder with inbuilt compass led us to slightly different plot corners than our GPS unit provided. In the field we adjusted our approach by depending on the rangefinder having believed that method to be more accurate.

We have revised our description accordingly. In addition, your insightful comment led us to remark on the instrumentation accuracy for all of our remote measurements.

“In each plot we first established a southwestern origin and measured its projected geographic coordinates (UTM Zone 12N in western plots and 13N in the Front Range) with a Garmin® eTrex 20 handheld GPS unit. Because the Garmin® eTrex units can have a positional error of up to 12 m, we opted to map trees and stumps on a Cartesian coordinate system rather than record geographic coordinates. Using the southwestern origin as our anchoring point, we then used Trimble® LaserAce 1000 laser rangefinders to divide the plot into four quadrants each 100 m x 100 m. The rangefinders had an inbuilt inclinometer to ensure that ranged distances accounted for topographic slope. The rangefinders’ lasers had a distance accuracy of 0.1 m, a compass heading accuracy of 2°, and an inclinometer accuracy of 2°. Once the approximate corners of quadrants were located we used three rangefinders to adjust corner locations to minimize positional error. By simultaneously measuring azimuth and distance from three positions to each corner in turn, we were able to minimize disagreement between individual rangefinders. This also ensured that quadrants maintained a square rather rhomboid shape. After the quadrants were laid out, we established a finer grid by dividing quadrants into 25 m x 25 m cells. Again, we used the same process of averaging the grid’s intersections with multiple rangefinders. Once each single grid cell was surveyed, its boundaries were considered immutable.”

Reviewer: Line 43: "a century of fire cessation" is probably too strong, absolute, and misleading. Fires occurred. It has been a century of actively controlling and stopping wildfires.

Authors: We agree and have changed this line to “In many of these forests, restoration is a common ecological objective as a century of fire suppression efforts has often yielded forest conditions which significantly deviate from historic forests”.

Reviewer: Line 45: Again, a statement probably too absolute ..."some forest managers now incorporate spatially..."

Authors: We agree and have revised our statement: “Made aware of the ecological significance of tree spatial patterning by Larson and Churchill, forest managers sometimes now incorporate spatially-explicit cutting guidelines into forest restoration plans.”

Reviewer: Lines 68-71: Besides the ecological research uses of stem maps, they can also be used in planning uneven-aged harvests, as is suggested in this Forests paper: https://doi.org/10.3390/f6041121.

Authors: We agree and have revised our statement: “Stem-maps of real-world plants can be used to investigate inter- and intra-specific interactions, hypothetical scenarios such as plant pest vulnerability, and may be used to inform silvicultural prescription development.”

Reviewer: Line 78: define "tibble".

Authors:  We clarified tibble: “Tibbles are a class structured as a data frame. A data frame is a heterogeneous, two-dimensional array with named columns; heterogenous means columns can be of any data type (i.e., numeric, character, logical, or complex).”

Reviewer: Line 153: Typo in the paper title.

Authors:  Typo fixed, thank you.

Reviewer: Line 156: What is "Cop." ?

Authors: Thank you for noticing this artefact of our citation management software. We verified that both ‘Ecography (Cop.)’ and ‘Ecography’ are used in journal catalogs to refer to that specific journal. We changed the citation’s journal title to ‘Ecography’ but will ultimately defer to MDPI Data’s copyediting decision.

Reviewer: Line 192: Inconsistent representation of authors' names.

Authors: Thank you for catching this error. We changed the citation to:

Churchill, D.J.; Larson, A.J.; Dahlgreen, M.C.; Franklin, J.F.; Hessburg, P.F.; Lutz, J.A. Restoring forest resilience: From reference spatial patterns to silvicultural prescriptions and monitoring. For. Ecol. Manage. 2013, 291, 442–457.